# Comment on Borșa et al. Developing New Diagnostic Tools Based on SERS Analysis of Filtered Salivary Samples for Oral Cancer Detection. *Int. J. Mol. Sci.* 2023, *24*, 12125

**DOI:** 10.3390/ijms252313030

**Published:** 2024-12-04

**Authors:** Ivan Bratchenko, Lyudmila Bratchenko

**Affiliations:** Laser and Biotechnical Systems Department, Samara National Research University, Moskovskoe Shosse 34, 443086 Samara, Russia; shamina94@inbox.ru

**Keywords:** liquid biopsy, SERS, cross-validation, PCA, limit of detection

## Abstract

This comment discusses a recent research paper on the classification of saliva samples with SERS by Borsa et al. The authors suggested utilizing PCA-LDA to detect oral cancer and claimed to achieve an accuracy of up to 77%. Despite the high prediction capability of the proposed approach, the demonstrated findings could be treated as unclear due to possible overestimation of the proposed classification models. Data should be provided for both the training and the validation sets to make sure that there were no repeated data from the same sample in either set. Moreover, the authors proposed to measure opiorphin in saliva with SERS as a potential biomarker of oral cancer. However, opiorphin in saliva is contained in ng/mL concentrations, and the proposed technique is most likely not capable of recording the real concentration of opiorphin.

Today, liquid biopsies based on spectral techniques [1,2] and especially on Raman spectroscopy [3,4] provide a way for the fast and precise analysis of body fluids [5,6,7]. The analysis of Raman spectra of tissues and biofluids can be utilized in a variety of applications such as cancer screening, non-communicable disease detection, virus tracking, etc. [8]. At the same time, the complexity and variability of spectral data and the even higher complexity of modern statistical approaches can be misleading and provide incorrect results [9,10,11,12].

Borsa et al. [13] examined saliva samples with SERS (surface-enhanced Raman spectroscopy) [14] to determine the possibility of oral cancer detection. To find the most important spectral areas and classify normal samples vs. cancerous ones, the authors utilized PCA-LDA (principal component analysis with linear discriminant analysis) [15]. The obtained accuracy in cancer detection was stated to be up to 77%, and opiorphin was proposed as a possible oral cancer biomarker. Basically, these methods are quite common in liquid biopsy; however, some details of the study indicate that the proposed classification models may have been overestimated and that opiorphin cannot be detected with the proposed approach in saliva in natural concentrations.

First, Borsa et al. [13] suggested using Leave-One-Out Cross-Validation (LOOCV) and compared the LOOCV accuracy with the training accuracy. This looks like an appropriate approach to validating classification models but, in conclusion to their study, the authors stated: “However, PCA-LDA yielded an overfit model (97% training accuracy vs. 83% Leave-One-Out Cross-Validation (LOOCV) accuracy). For that reason, we used five PCA components which account for 90% of the total variance and are less overfit in the PCA-LDA analysis (85% training accuracy vs. 77% LOOCV accuracy).” The authors’ choice of the principal components (PCA components) was based on the proximity of the results for training and LOOCV. However, this approach may have led to an inadequate estimation of the classification model results. For reference, we have provided Figure 1 with the data from our previous studies [16,17], where we compared the data used for training (blue line) and cross-validation (red line). As can be seen from the figure, a higher number of principal components (e.g., eight principal components) can yield slightly overfitted data, but the number of principal components should be limited in the first local minimum on a cross-validation curve. Otherwise, the utilized classification model will utilize junk components that may even decrease the performance of the model. Note here that high performance of eight and nine principal components is just a random result. Thus, the authors should provide the same data to prove the correct number of principal components. In addition, Borsa et al. could have analyzed a scree plot to choose the proper number of principal components [15]. Overall, for groups of about 17 samples, only 2–3 first principal components can be effectively utilized for constructing classification models [9].

Another important issue in the commented paper is the ability to detect opiorphin. Their conclusion stated the following: “The presence of opiorphin in salivary samples was experimentally proven for the first time”. The SERS spectra of opiorphin (Figure 2) were collected with a concentration of 0.1 mg/mL. However, in real saliva samples, the concentration of opiorphin can be in the range of about 3–30 ng/mL [18]. Therefore, to estimate the contribution of opiorphin nanograms (per ml) in real saliva samples, the authors should have reduced the opiorphin spectra in Figure 2 by a factor of at least 10,000 (about 10^4^ to 10^6^). In doing so, the opiorphin contribution to SERS spectra of saliva would decrease and may become as low as measurement noise. Note here that Borsa et al. associated the 1002 cm^−1^ band with the opiorphin presence; however, this spectral band may be associated with tens of substances that are present in saliva with higher concentrations (e.g., uric acid, exosomes [19,20,21], etc.).

In addition, it should be noted that certain molecules have higher affinities for metal surfaces, and therefore the SERS spectra of biofluids can be dominated, for example, by the contributions of purine (uric acid, hypoxathine, adenine) and sulfur-containing (ergothioneine, glutathione) metabolites [22,23]. Thus, to demonstrate the real concentration of an exact analyte, SERS measurements should be accompanied with other well-established techniques (such as liquid chromatography or other laboratory techniques) [4]. Without determining the exact concentration of the target analyte and the ability to detect this analyte in real concentrations with SERS, numerous biological molecules may be hypothetically treated as biomarkers due to their potential contribution to the recorded SERS spectra.

In summary, to prove the correctness of the classification models, it is advisable to provide additional data regarding the utilized cross-validation procedure. It is necessary to compare the performance of training and cross-validation and to estimate a scree plot. The authors should demonstrate that their technique is capable of detecting opiorphin in saliva samples at ng/mL concentrations (or to establish the limit of opiorphin detection).

## Figures and Tables

**Figure 1 ijms-25-13030-f001:**
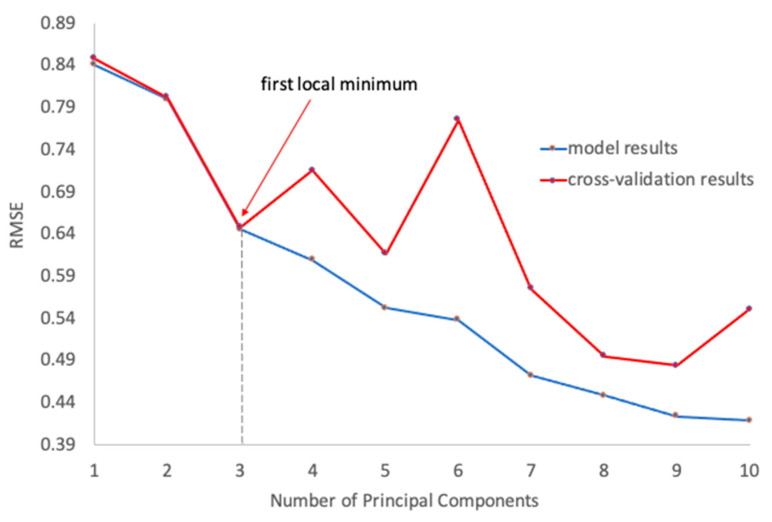
Example of utilizing RMSE for determining the number of PCs in the PLS-DA model for discriminating between a group of patients with kidney failure and a group of healthy volunteers in the analysis of human skin spectra (obtained with permission from the authors [17]).

**Figure 2 ijms-25-13030-f002:**
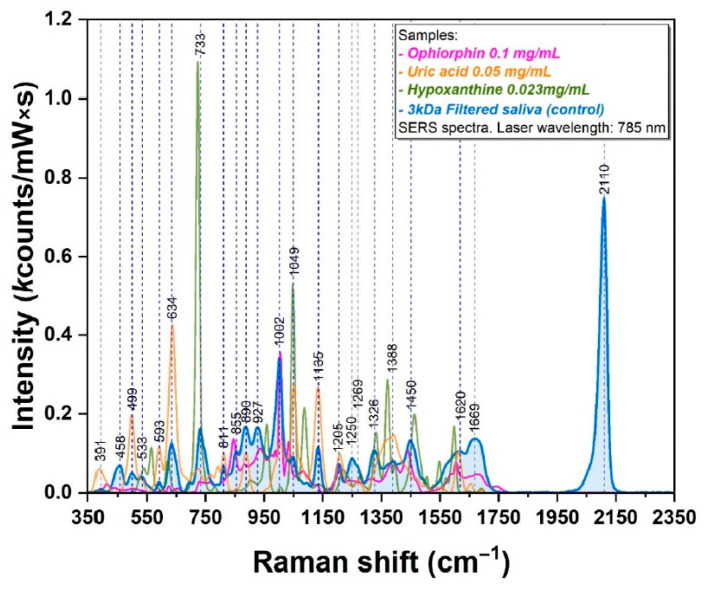
SERS spectrum of opiorphin, uric acid, hypoxanthine, and salivary control probes using the excitation wavelength of 785 nm (from Figure S5 in the commented paper [13]).

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
