# Peer review of "Comment on Borșa et al. Developing New Diagnostic Tools Based on SERS Analysis of Filtered Salivary Samples for Oral Cancer Detection. Int. J. Mol. Sci. 2023, 24, 12125"

_ijms, 2024, doi:10.3390/ijms252313030_

Round 1

Reviewer 1 Report

Comments and Suggestions for Authors

The authors have commneted previously publication on oral cancer detection using SERS. Although this is a topical and timely topic, there are a few issues that need to be addressed before suggesting for publication. The use of PCA is commonly used multi variate analysis, aiming to dricrminate between data sents – in this case - aruired raman scpetra. The authors have validated their proposed Raman/SERS strategy on patient derived samples and uaed healthy conterpart for control. Althugh the diffrence they have noticed is rather limited to the peaks’ intensity rather than position, this could be explained as aboundabce of targeted biomarker is quite low. 

The abstract needs to be completely revised as the language and style of writing is rather confusing.

The language tone used in this paper is passive aggressive towards Borsa et al, and it is suggested to tone it down and try to be constructive.

It is recommended that the authors provide more references in the introduction to further strengthen the statements. 

It is suggested that the authors provide other examples of insufficiency of opiorphin as a biomarker in saliva reported by other researchers or some other evidence.

Comments on the Quality of English Language

Extensive editing of English language required

Author Response

Comment 1:

The abstract needs to be completely revised as the language and style of writing is rather confusing.

Reply:

We added an additional sentence to the abstract to highlight the description of specific issues discussed in the comment. “The data should be provided both for the training set and for the validation set to make sure that there is no repeated data from the same sample in the training and validation sets.” The text of a comment was checked by a native speaker.

Comment 2:

The language tone used in this paper is passive aggressive towards Borsa et al, and it is suggested to tone it down and try to be constructive.

Reply:

Language tone was softened. The idea of the comment is to point out possible drawbacks of the original article in a constructive manner.

Comment 3:

It is recommended that the authors provide more references in the introduction to further strengthen the statements. 

Reply:

We added additional references (e.g. from Bonifacio et al) to strengthen the statements in the comment.

Comment 4:

It is suggested that the authors provide other examples of insufficiency of opiorphin as a biomarker in saliva reported by other researchers or some other evidence.

Reply:

In the text of the comment we do not claim that opiorphin may not be used as saliva biomarker. It seems like it may be a biomarker of some diseases (e.g. doi.org/10.1155/2021/3639441). But as we describe in the text, in the approach presented in the commented paper, opiorphin detection is impossible with natural concentrations. Thus, it is overoptimistic to expect that SERS-based approach may utilize opiorphin in saliva as biomarker.

Reviewer 2 Report

Comments and Suggestions for Authors

In the excerpt provided, the authors discuss the use of liquid biopsy techniques, specifically Raman spectroscopy, for the analysis of body fluids. Based on the results of a study by Borsa et al, the authors comment on the use of surface-enhanced Raman spectroscopy (SERS) in saliva samples to detect oral cancer. The Borsa et al, used principal component analysis with linear discriminant analysis (PCA-LDA) to identify and differentiate between normal and cancerous spectral areas. They achieved a 77% accuracy rate in cancer detection and suggested the biomarker opiorphin for oral cancer. Although the proposed methods align with common liquid biopsy approaches, the details of the study indicate possible overestimation of the classification models and the inability to detect opiorphin in saliva at natural concentrations. The authors critique the use of Leave-One-Out Cross-Validation (LOOCV) and the selection of principal components based on the proximity of training and LOOCV results. They argue that this approach may lead to an inadequate estimation of classification model results. Instead, they recommend limiting the number of principal components to the first local minima on the cross-validation curve, as utilizing additional components may introduce junk components that decrease model performance. The authors also suggest the examination of a scree plot for selecting the appropriate number of principal components. In summary, the authors suggest that while the proposed classification models have potential, its estimation may be flawed. They emphasize the importance of choosing the correct number of principal components for effective model construction, ideally limiting it to 2-3 components for groups of approximately 17 samples. Bratchenko et al. has made the clear point of view of their understanding of utilizing the higher Principle component 8 can provide overfitting of data. 

Author Response

Comment 1:

In the excerpt provided, the authors discuss the use of liquid biopsy techniques, specifically Raman spectroscopy, for the analysis of body fluids. Based on the results of a study by Borsa et al, the authors comment on the use of surface-enhanced Raman spectroscopy (SERS) in saliva samples to detect oral cancer. The Borsa et al, used principal component analysis with linear discriminant analysis (PCA-LDA) to identify and differentiate between normal and cancerous spectral areas. They achieved a 77% accuracy rate in cancer detection and suggested the biomarker opiorphin for oral cancer. Although the proposed methods align with common liquid biopsy approaches, the details of the study indicate possible overestimation of the classification models and the inability to detect opiorphin in saliva at natural concentrations. The authors critique the use of Leave-One-Out Cross-Validation (LOOCV) and the selection of principal components based on the proximity of training and LOOCV results. They argue that this approach may lead to an inadequate estimation of classification model results. Instead, they recommend limiting the number of principal components to the first local minima on the cross-validation curve, as utilizing additional components may introduce junk components that decrease model performance. The authors also suggest the examination of a scree plot for selecting the appropriate number of principal components. In summary, the authors suggest that while the proposed classification models have potential, its estimation may be flawed. They emphasize the importance of choosing the correct number of principal components for effective model construction, ideally limiting it to 2-3 components for groups of approximately 17 samples. Bratchenko et al. has made the clear point of view of their understanding of utilizing the higher Principle component 8 can provide overfitting of data. 

Reply:

We thank the reviwer for the feed back of the comment.

According to other reviewers comments we added to the text additional explanations and references to strengthen the statements in the comment. 

The text of a comment was checked by a native speaker.

Round 2

Reviewer 1 Report

Comments and Suggestions for Authors

The revised manuscript is suggested for publication.